# OPENLEAF: OPEN-DOMAIN INTERLEAVED IMAGE-TEXT GENERATION AND EVALUATION

## ABSTRACT

This work investigates a challenging task named open-domain interleaved image-text generation, which generates interleaved texts and images following an input query. We propose a new interleaved generation framework based on prompting large-language models (LLMs) and pre-trained text-to-image (T2I) models, namely *OpenLEAF*. In *OpenLEAF*, the LLM generates textual descriptions, coordinates T2I models, creates visual prompts for generating images, and incorporates global contexts into the T2I models. This global context improves the entity and style consistencies of images in the interleaved generation. For model assessment, we first propose to use large multi-modal models (LMMs) to evaluate the entity and style consistencies of open-domain interleaved image-text sequences. According to the LMM evaluation on our constructed evaluation set, the proposed interleaved generation framework can generate high-quality image-text content for various domains and applications, such as how-to question answering, storytelling, graphical story rewriting, and webpage/poster generation tasks. Moreover, we validate the effectiveness of the proposed LMM evaluation technique with human assessment. We hope our proposed framework, benchmark, and LMM evaluation could help establish the intriguing interleaved image-text generation task.

## 1 INTRODUCTION

This work investigates an intriguing yet challenging task, namely open-domain interleaved image-text generation. As shown in Fig. 1, given an arbitrary open-domain query as the input, the task aims to generate a sequence of interleaved text descriptions and illustration images to form a coherent content following the input query. The ultimate goal of open-domain interleaved generation is to seamlessly generate arbitrary multi-modal contents, thus facilitating a wide range of applications and functionalities, such as generating multi-modal illustrations and web pages, storytelling, chain-of-thought explanations, and so on.

Early explorations (Li et al., 2019; Zeng et al., 2019; Li et al., 2020; Song et al., 2020; Maharana et al., 2021; Maharana & Bansal, 2021; Maharana et al., 2022; Szűcs & Al-Shouha, 2022; Pan et al., 2022; Liu et al., 2023a) simplify interleaved generation by narrowing down the problem to specific sub-domains, such as the story telling as shown in Fig. 1. Despite the promising explorations, these methods can only generate story illustrations in a format in which each image is paired with one single sentence. Therefore, these methods cannot achieve open-domain interleaved generation with arbitrarily interleaved content (Zhu et al., 2023; Gadre et al., 2023; Laurençon et al., 2023), where image and text are organized in diverse interleaved formats for input instructions covering a wide range of open-domain topics.

For the interleaved content evaluation, prior studies typically train a distinct evaluator for each aspect within a specific domain, such as a story character recognizer (Li et al., 2019; Maharana et al., 2021) for character consistency in story generation. However, these isolated evaluators cannot scale to open-domain interleaved generation due to the difficulty in generalizing to open-domain generative scenarios. Consequently, the remaining challenges of the open-domain interleaved generation are the lack of a proper evaluation approach for varying topics and diverse formats, a solid baseline, and a benchmark dataset to compare different methods on.

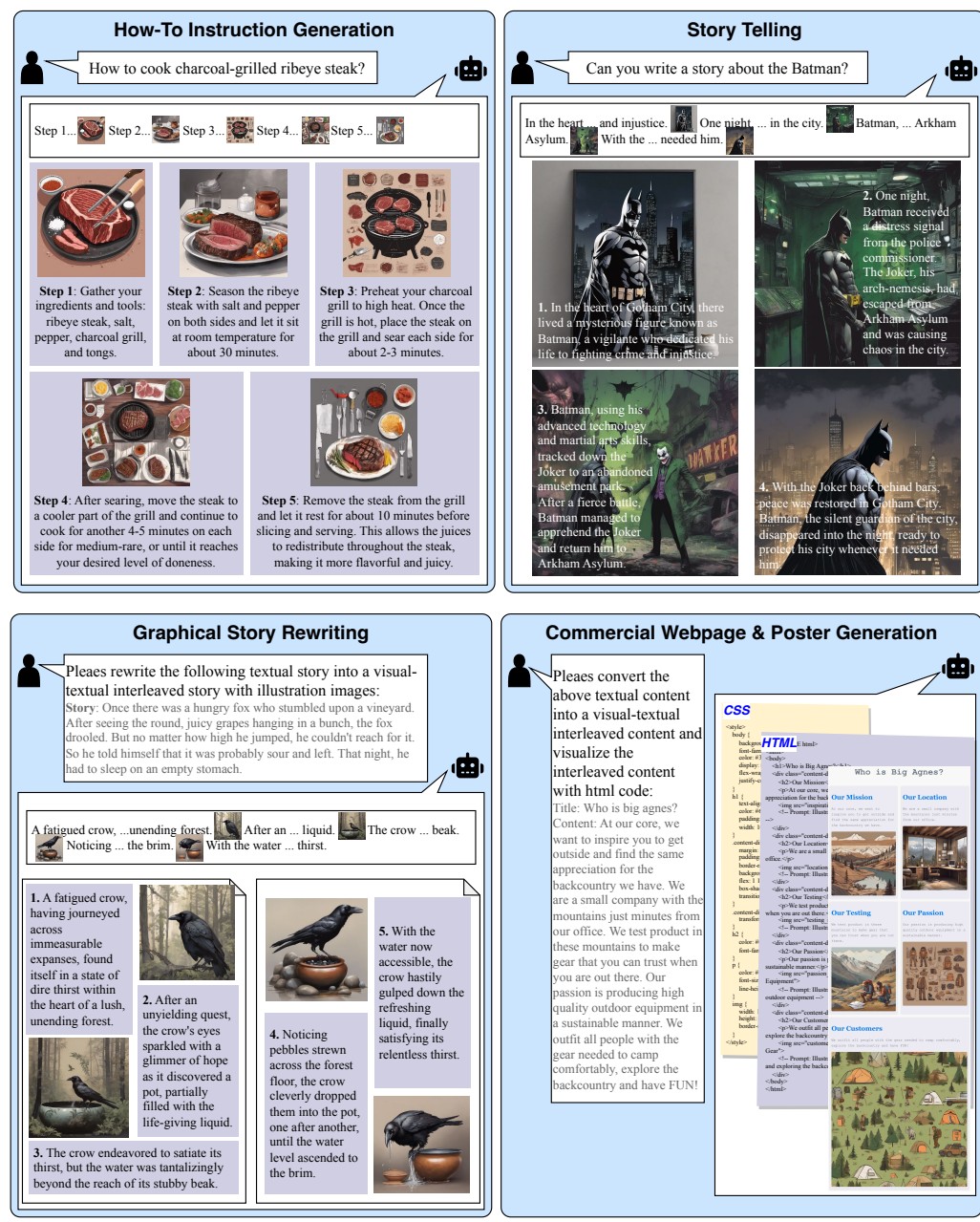

Figure 1: Examples of open-domain interleaved content generation. We show baseline results on producing visual how-to instructions (top-left), generating multi-modal stories (top-right), converting textual stories to multi-modal stories (bottom-left), and generating webpages and posters via HTML and CSS codes (bottom-right).

To address the lack of a baseline model, we propose a new training-free open-domain interleaved image-text generation framework based on GPT-4 (OpenAI, 2023b) and Stable Diffusion XL (SDXL) (Podell et al., 2023), named *OpenLEAF*. Given an arbitrary user query, *OpenLEAF* first prompts GPT-4 to generate a sequence of textual descriptions including image placeholders. Next we use GPT-4 to generate T2I prompts that fit well in the interleaved content. To improve the entity and style consistency of images within the interleaved content, we add global entity and style contexts to T2I prompts, where the entity context is short appearance descriptions of the common subjects throughout the interleaved content while the style context defines the target image style, both are produced by prompting GPT-4. Finally, SDXL converts T2I prompts into images, completing the interleaved content by replacing each image tag with the corresponding image.

For the interleaved content evaluation, inspired by the power of recent large multi-modal models (LMMs) (OpenAI, 2023b; Google, 2023; Microsoft, 2023), we explore prompting LMMs for unified interleaved content evaluation. Specifically, We use BingChat for the evaluation due to the empirical observation that it performs the best in assessing the quality of an interleaved image-text sequence from different perspectives. Ideally, an interleaved sequence should contain relevant and informative content, maintain consistency in subject identity and appearance, and feature a coherent image style. Therefore, we examine the entity and style consistency of the generated sequence, each broken down into sub-topics for LMM assessment. We then aggregate the generated scores as the final rating.

We collect a benchmark dataset for evaluating open-domain interleaved generation methods, which consists of 30 input queries, covering a wide range of topics and formats, such as visual instruction generation, story generation and rewriting, webpage, and poster generation. The experiments on this new benchmark based on the LMM-based evaluation approach demonstrate that the proposed baseline method can produce good interleaved results, achieving improvement against a simplified variant. In addition, we design comprehensive human evaluation and LMM analysis on the benchmark to validate the effectiveness of the BingChat evaluation.

Our contributions are summarized as follows.

- We study the open-domain interleaved image-text generation task, which aims to synthesize open-domain interleaved image-text sequences with arbitrary formats.
- We first explore LMM's assessment ability to evaluate interleaved image-text sequences. Comprehensive human evaluation and analysis validate the effectiveness of the LMM-based evaluation technique.
- We propose *OpenLEAF* framework as a strong baseline for the interleaved image-text generation task, which improves the semantic consistency and style consistency with the proposed global context. We benchmark *OpenLEAF* on the developed evaluation set that covers diverse topics and sequence formats.

## 2 RELATED WORK

**Interleaved Image-Text Generation.** Starting from the GAN-based methods (Li et al., 2019; Zeng et al., 2019; Li et al., 2020; Song et al., 2020; Maharana et al., 2021; Maharana & Bansal, 2021; Szűcs & Al-Shouha, 2022), current methods that address the interleaved image-text generation task include StoryGen (Liu et al., 2023a), AR-LDM (Pan et al., 2022), and StoryDALL-E (Maharana et al., 2022), where Liu et al. (2023a) and Pan et al. (2022) fine-tune latent diffusion models (LDMs) (Rombach et al., 2022) on sequential story-like images, leading to auto-regressive LDMs, while Maharana et al. (2022) fine-tunes a text-to-image transformer. However, all of the above methods cannot be applied to the open-domain interleaved generation since the fine-tuning process makes the above model only work well on images similar to the fine-tuning datasets, leading to limited generation domains. On the other hand, how to evaluate open-domain interleaved content remains to be unsettled.

Recent multi-modal LLMs such as GILL (Koh et al., 2023), Emu (Sun et al., 2023), and Dream-LLM (Dong et al., 2023) show the decent performance on open-domain image-text generations and perceptions. However, they are not specifically designed for interleaved generation and evaluation, leaving the open-domain interleaved generation task unestablished.

**Foundation Models for Open-Domain Evaluation.** How to evaluate open-domain content has drawn increasing attention. In the natural language process, studies show promises of prompting LLMs such as GPT for open-ended text evalaution (Chiang & Lee, 2023; Liu et al., 2023c; Fu et al., 2023). For evaluating the visual-language content, CLIPscore (Radford et al., 2021), VisualGPTScore (Lin et al., 2023), and LLaVA-based scoring methods (Black et al., 2023; Liu et al., 2023b) effectively evaluates the open-domain image-text similarity. However, these methods can only work on single image-text pairs while having limited capabilities to evaluate arbitrarily interleaved content comprehensively.

This work uses LMMs (*i.e.*, BingChat (Microsoft, 2023)) for evaluating the open-domain interleaved content, which addresses the above-mentioned issue by accepting multiple image-text pairs and allowing for open-ended evaluation.

**Multi-Modal Agents.** The method part of this paper is related to multi-modal agent studies (Gupta & Kembhavi, 2023; Surís et al., 2023; Wu et al., 2023; Yang* et al., 2023; Shen et al., 2023; Li et al., 2023), which chain LLMs with multi-modal tools for new tasks. For example, Visual ChatGPT (Wu et al., 2023) shows that allocating various generative models (Rombach et al., 2022; Meng et al., 2021; Zhang & Agrawala, 2023) with ChatGPT (OpenAI, 2023a) can achieve complicated image generation and editing. Differently, our work focuses on a specific challenging task of open-domain interleaved image-text generation.

## 3 METHOD

This section introduces *OpenLEAF* and the evaluation pipeline based on LMM. Fig. 2 overviews our method. Subsequently, we detail our generation and evaluation pipelines, respectively.

### 3.1 INTERLEAVED CONTENT GENERATION

We achieve open-domain interleaved generation based on GPT-4 (OpenAI, 2023b) and SDXL (Podell et al., 2023). The top panel of Fig. 2a shows the generation pipeline of our method. Given an arbitrary user query, we initially follow a meticulously designed composition strategy to assemble an input prompt that indicates the content, format, and constraints of the target output. We then feed the input prompt into GPT-4, which generates the textual descriptions, determines the positions to insert images, and formulates the visual prompt for each image. Subsequently, we incorporate global entity and style contexts into the visual prompts to improve the entity and style consistencies of SDXL. Here, the entity context comprises the appearance descriptions of common subjects, while the style context is a unique image style description shared across all visual prompts. Finally, SDXL converts visual prompts into real images, thereby creating the interleaved content.

**User Query Composition.** The input prompts to GPT-4 consist of four parts. We first add a few in-context examples to the start of the prompt. Each example shows the desired output corresponding to a specific input query. The in-context examples enable GPT-4 to comprehend the expected content more effectively and encourage it to generate content in the format of the in-context examples, facilitating easier automatic extraction of results. Subsequently, we concatenate the generation instruction with the user input to form the prompt. In this case, the instruction tells GPT-4 the desired output type, while the user input specifies the detailed content. Finally, we add a few control sentences to the prompt to control the number of image placeholders, story sentences, instruction steps, and `<div>`s in HTML, *etc*.

**Text Generation.** The first step of *OpenLEAF* is to generate text. By feeding the assembled prompt discussed in the previous part, we enable GPT-4 to produce all text descriptions and image placeholders, indicating the position of each image. For example, as shown in the text generation panel of Fig. 2a, when generating stories and how-to instructions, GPT-4 is prompted to generate story sentences and instructional steps, respectively, where image tags `` is also included in the generated text. Each image tag indicates the position of the corresponding image, forming an initial interleaved structure. When generating HTML code, the position of each image is determined by the placement of the `` environment, where the generated CSS code can further tune the size, position, and alignment of each image. Next, we prompt GPT-4 to generate visual prompts from text descriptions. In this step, the input prompt also follows the composition strategy introduced earlier, incorporating all story sentences or instructional steps into the user input part. This approach allows GPT-4 to capture the context of the whole story or how-to instructions when generating the visual prompt for each image.

**Adding Global Context.** To improve the entity and style consistencies of images within the interleaved content, we introduce global entity and style context into the visual prompts before feeding it into SDXL to generate images. For the global entity context, we add a short appearance description of each common subject to the visual prompts, where GPT-4 is used to extract common subjects from text content, generate appearance descriptions, and rewrite visual prompts. To improve the style consistency of images, we prompt GPT-4 to determine a proper visual style to depict the interleaved content, based on the generated text descriptions. For example, GPT-4 indicates that a vibrant color palette and comic book style are best to illustrate superhero stories. Then a short image style description is added to the beginning of each visual prompt to control the artistic style of images

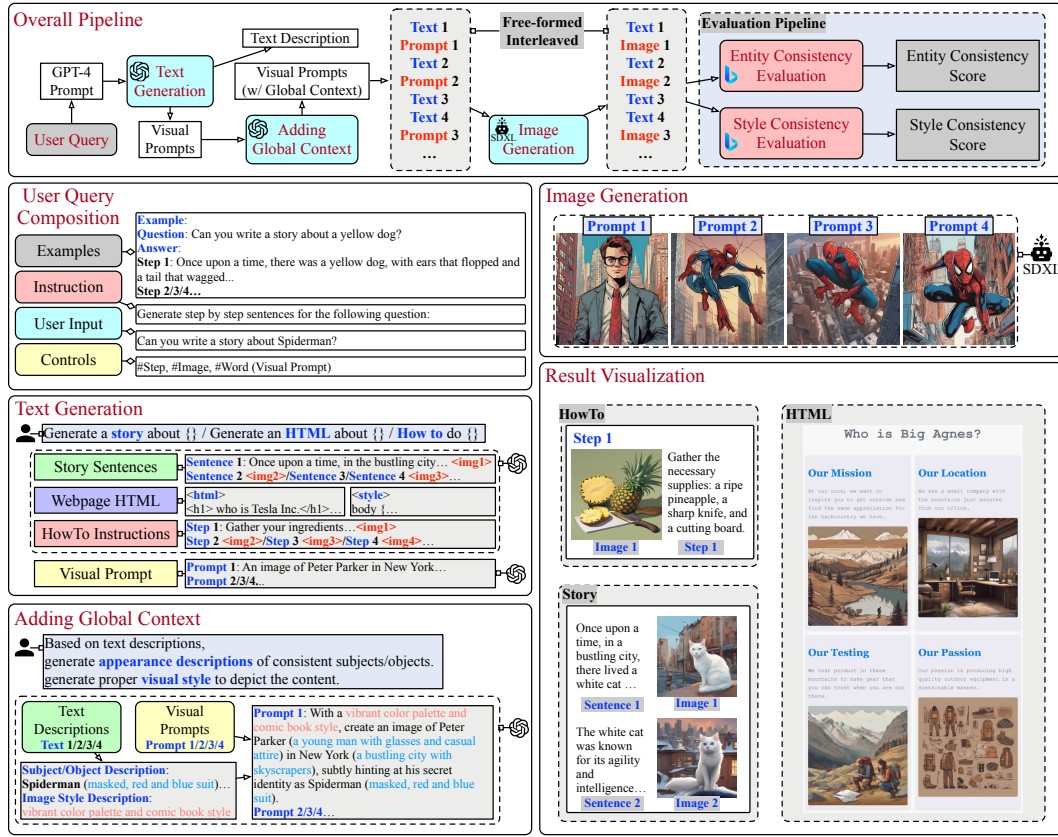

(a) Our interleaved generation framework. The top panel illustrates the overall framework while other panels show details of each procedure.

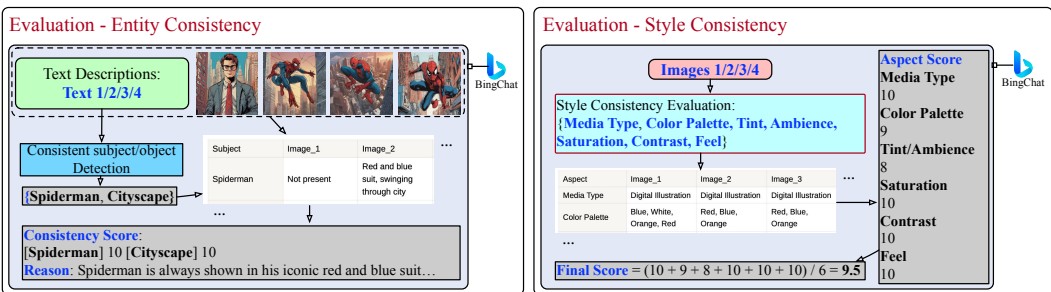

(b) Our LMM-based interleaved evaluation pipeline.

Figure 2: Overviews of the proposed interleaved generation framework: (a) and LMM-based evaluation pipeline (b).

generated by SDXL. Fig. 2a shows the process of adding global context into visual prompts. The visual prompts equipped with the global context are then converted into images by SDXL, resulting in the interleaved content.

## 3.2 LMM-BASED EVALUATION.

We use BingChat (Microsoft, 2023) to evaluate the quality of interleaved content based on the chain-of-thought approach, with a focus on the entity and style consistencies.

**Entity Consistency Evaluation.** To evaluate the entity consistency of images within the interleaved content. We first prompt BingChat to detect two main common subjects from the generated text descriptions. For example, in Fig. 2b, the Spiderman and the background cityscape are the main subjects in images, which should have consistent entities and appearances between images. Subse-

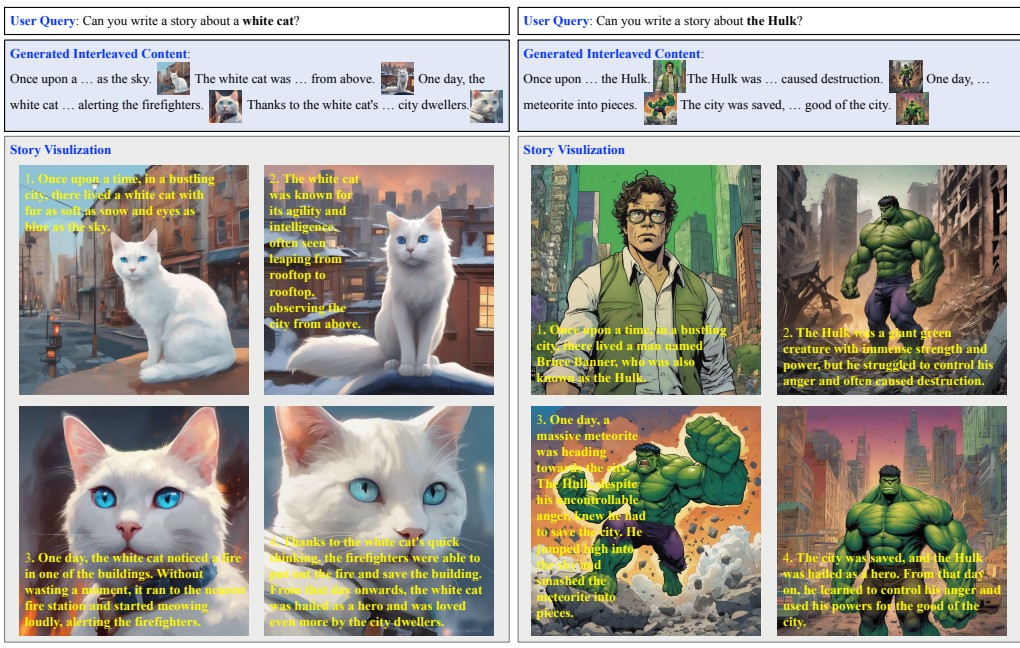

(a) Generating multi-modal stories.

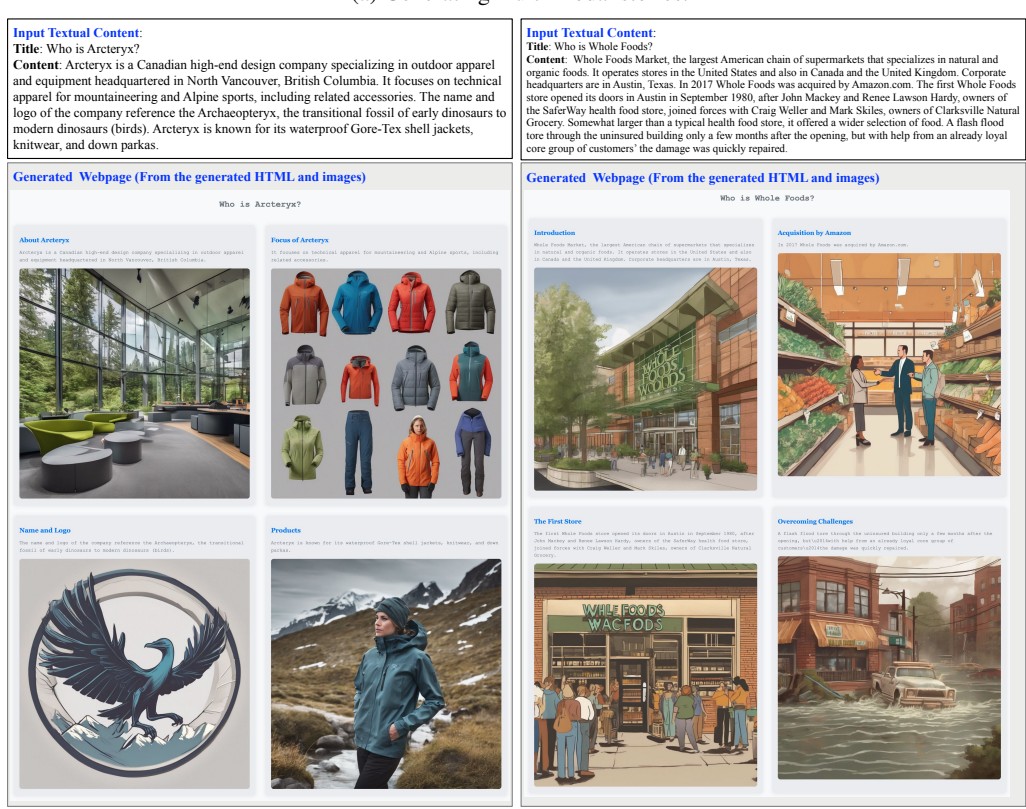

(b) Converting text introductions into graphical webpages.

quently, we input all images into BingChat, assigning a unique index `image_{i}` to each image. We then have BingChat summarize the appearance of each common subject in the images and assign a score for the entity consistency based on the appearances of common subjects.

**Style Consistency Evaluation** We evaluate the style consistency of images based on seven visual factors: media type, color palette, tint, ambiance, saturation, contrast, and overall feel of images.

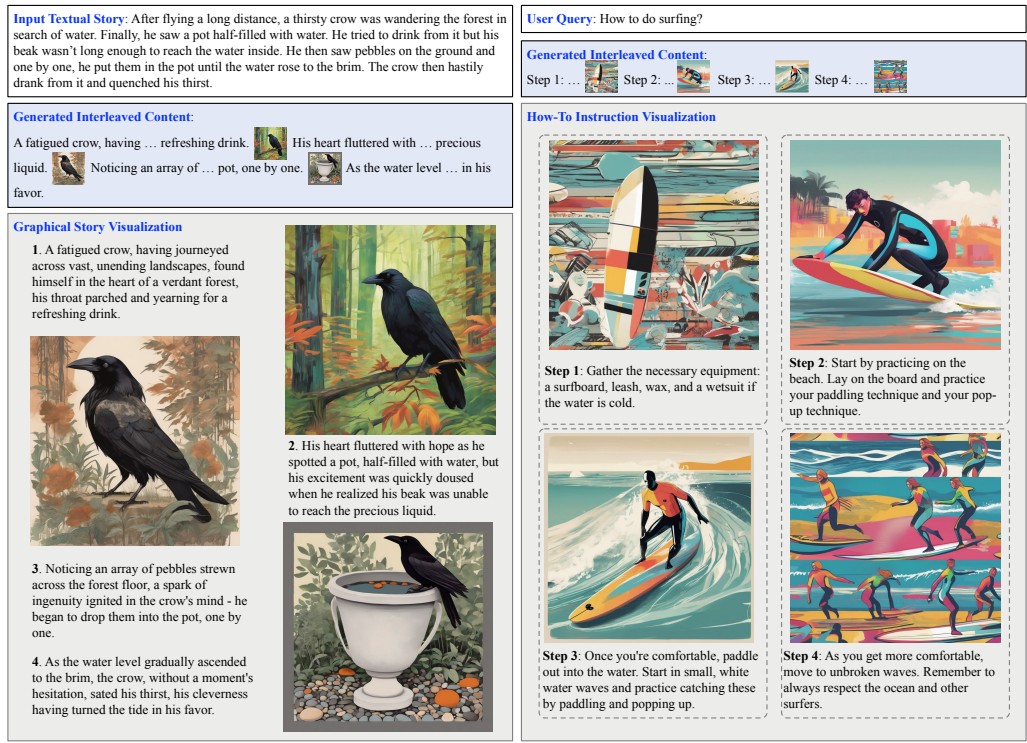

(c) Converting a text story into a multi-modal story.      (d) Generating visual how-to instructions.

Figure 3: Interleaved visual-language generation results of *OpenLEAF*. For story generation, story rewriting, and how-to instruction generation results, we show the raw interleaved content below the user query. Please zoom in on screen to see details.

The media type and feel recognize the global feeling of an image such as a realistic photo, painting, digital illustration, or cartoon. While the other evaluation factors detect more subtle visual style distinctions within images. Similar to the entity evaluation, we use BingChat to summarize the visual style of each image based on the pre-defined visual factors and score the style consistency for each factor. The final score is calculated as the mean value of the consistency score on all visual factors.

## 4 EXPERIMENTS

This section introduces our experimental settings, results, and analysis.

### 4.1 EXPERIMENT SETTINGS

**Implementation Details.** We utilize the text-only GPT-4 API of July 2023. For T2I generation, we use the text-to-image pipeline of the SDXL v1.0 model [1], with all hyper-parameters set to their defaults. For LMM-Evaluation, we apply the precise mode of BingChat to both entity and style consistency evaluations. We develop a Python code to connect all the above prompting procedures, which automatically extracts the desired content from the previous model's output and forms the input for the next procedure.

**Compared Model Variants.** To demonstrate the effectiveness of the global context, we compare the performance of the proposed baseline and a simplified variant without the global context.

**Evaluation Set.** We collect a benchmark dataset of thirty problems, which covers four interleaved generation tasks: graphical storytelling, visual how-to instruction generation, text-to-graphic story

---
[1] https://huggingface.co/stabilityai/stable-diffusion-xl-base-1.0

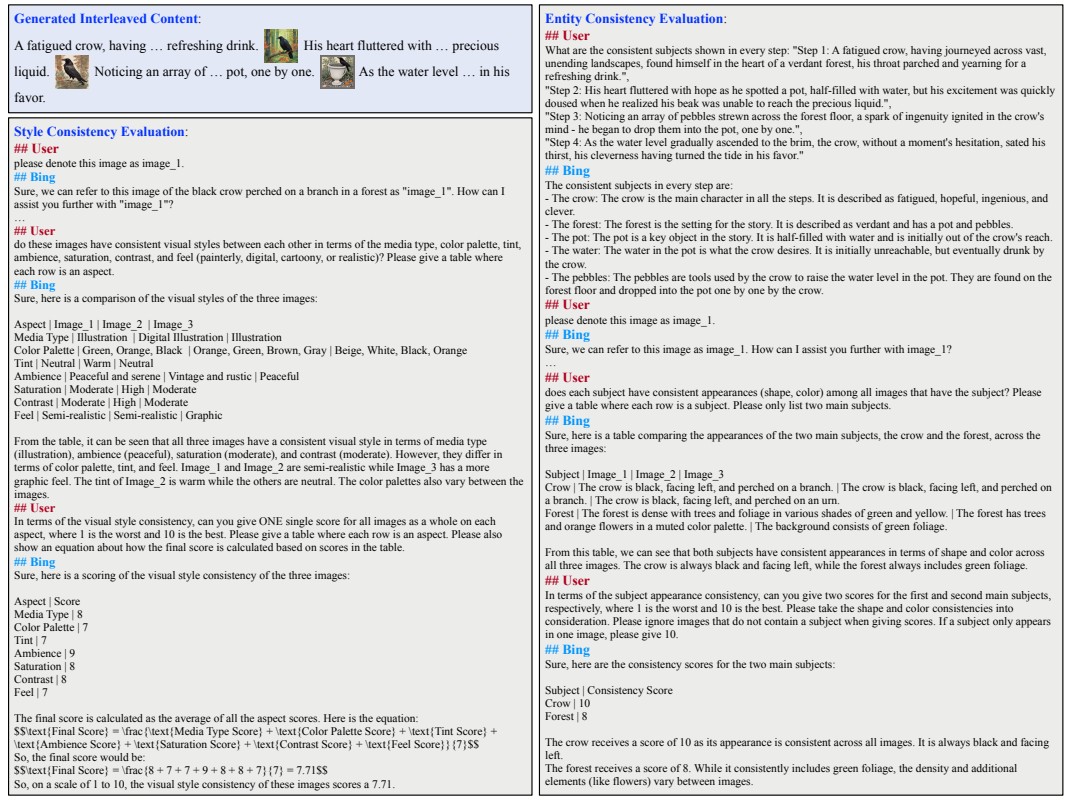

Figure 4: The entity and style consistency evaluation output of BingChat. BingChat can effectively generate rational responses, understand the meaning of pre-defined evaluation aspects, and give scores based on evidence.

rewriting, and webpage generation. The how-to instruction generation task has ten problems, which include the requests to generate the recipes for foods and the tutorials for sports and actions. The webpage generation task has five questions about converting a company introduction into a webpage or poster. The ten storytelling and five graphical story rewriting problems ask the model to generate animal- or superhero-related stories. Please refer to the supplementary material for problems of the constructed benchmark.

## 4.2 EXPERIMENT RESULTS

**Qualitative Results.** Fig. 3 shows the qualitative results of the *OpenLEAF*. We observe that *Open-LEAF* generates coherent interleaved image-text sequences based on arbitrary input queries. For example, in Fig. 3 (a), *OpenLEAF* generates coherent and interesting multi-modal stories about a white cat and the Hulk. All images have consistent entities and styles while being visually appealing. Fig. 3 (b) shows *OpenLEAF* generating commercial webpages based on introductions of "Arcteryx" and "Whole Foods". *OpenLEAF* naturally splits text introductions into paragraphs and accurately adds paragraph headlines. The generated HTML and CSS files arrange the interleaved content in an intuitive and attractive way. Fig. 3 (c) shows *OpenLEAF* converting a textual story into multi-modal story and generating visual how-to instruction. The story sentences in the graphical story become more vivid compared with the input textual story while visually appealing and coherent images are naturally inserted into arbitrary text locations. The generated visual how-to instructions are informative and the illustration images are helpful in understanding terminologies of surfing.

More importantly, Fig. 4 shows that the BingChat is effective in evaluating entity and style consistencies, even for the complicated interleaved image-text sequences. For example, in evaluating entity consistency, BingChat accurately grounds common subjects extracted from text descriptions to the corresponding subjects in images. We observe that BingChat are capable of generating rationale text, understanding the meaning of and giving consistency scores on pre-defined sub-categories, as well as calculating the final score based on a clear mathematical equation.

Table 1: The mean and variance of the BingChat evaluation on the benchmark dataset. Adding global context improves the averaged consistencies and lowers the variances.

| Model | Entity Consistency | | Style Consistency | |
|---|---|---|---|---|
| | mean↑ | variance↓ | mean↑ | variance↓ |
| Ours w/o Global Context | 7.84 | 1.22 | 8.00 | 0.88 |
| Ours w/ Global Context | **8.40** | **0.77** | **8.22** | **0.79** |

Table 2: The correlation analysis between the human and BingChat evaluations. $p$ denotes the p-value.

| Correlation Index | Entity Score | Style Score |
|---|---|---|
| Kendall's Tau↑ $\in (-1, 1)$ | $0.87\ (p = 0.0008)$ | $0.58\ (p = 0.0196)$ |
| Spearman's Correlation↑ $\in (-1, 1)$ | $0.78\ (p = 0.0080)$ | $0.95\ (p = 0.0000)$ |

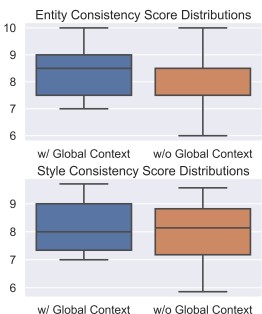

Figure 5: The distribution comparison of BingChat evaluation scores.

**Quantitative Results.** We then evaluate *OpenLEAF* and its variants using the effective BingChat evaluation. We compute the entity and style consistency scores on the interleaved content generated from problems in the benchmark set. Table 1 shows the mean and variance values of the entity and style consistency scores over the benchmark set while the boxplots in Fig. 5 show the comparison of the score distributions. Both *OpenLEAF* and its variant show good entity and style consistencies. After adding the global context, we observe that the mean values of both entity and style consistency scores increased, indicating improved consistencies. On the other hand, the global context also reduced the variances of both entity and style consistency scores. demonstrating that adding global context can improve the robustness of *OpenLEAF* in terms of content and style consistencies.

### 4.3 EVALUATION PIPELINE ANALYSIS

To ensure that the LLM-evaluation pipeline can produce the entity and style consistency scores in line with the human perception. We conduct an analysis to study the score correlation between LLM-evaluation and the human rating. We collect ten image pairs from our generated interleaved content, where each pair comes from the same story. First, we use LLM-evaluation to give both entity and style consistency scores on each image pair. Next, we conduct a user study to let users rate the entity and style consistencies of each image pair. Considering that humans may have difficulty giving an absolute score of consistency, in the user study, we randomly select two image pairs from the evaluation set and show them side-by-side to users, letting users choose one pair that has better entity or style consistency. Finally, we calculate the consistency scores of each image pair based on the number of user preferences.

To determine whether the proposed LLM-evaluation aligns well with human perception, we compute Kendall's Tau (Kendall, 1938) and Spearman's Correlation (Kendall & Stuart, 1973) indexes, which measure the similarity between the rankings from the human and LLM-evaluation scores. Table 2 shows the correlation scores. For both entity and style consistencies, we achieve Kendall's Tau and Spearman Rank-Order close to 1 with low p-values, strongly indicating that our LLM-evaluation method has a good alignment with the human evaluation in both measuring entity and style consistencies of the interleaved content.

## 5 CONCLUSION

In this paper, we fill the gap of open-domain interleaved image-text generation by introducing a baseline interleaved generation method *OpenLEAF* based on GPT-4 and SDXL, an evaluation pipeline based on BingChat, and a benchmark dataset to compare different approaches. Experimental results on the constructed benchmark dataset show that *OpenLEAF* has a strong ability to generate arbitrarily interleaved image-text sequences for addressing open-domain user queries. A comprehensive analysis based on the user study demonstrates that the evaluation method based on BingChat can effectively capture the styles and entities within images, thereby reliably evaluating generated interleaved multimodal contents.

**Ethics Statement.** In this research on the open-domain interleaved image-text generation, we uphold a commitment to ethical conduct guided by principles that prioritize human welfare, privacy, fairness, accountability, and transparency. Our model is build on top of GPT-4, BingChat, and Stable Diffusion XL, where all of the pre-trained models are publicly available and we believe their owners have a clear awareness of addressing the potential ethics issues. Our method does not bring new ethics issues. On the data side, we do not include any human-related problems into our constructed interleaved benchmark dataset to avoid the potential privacy and ethics issues.

**Reproducibility Statement.** All the pre-trained models we used are publicly available, where we clearly indicate the model version, settings, and hyper-parameters we used in our experiments. We believe our method and the evaluation pipeline are reproducible.

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
