# SUPPLEMENTARY MATERIAL OF OPENLEAF: OPEN-DOMAIN INTERLEAVED IMAGE-TEXT GENERATION AND EVALUATION

## A  BENCHMARK PROBLEMS

Table 1, 2, and 3 show the complete list of problems in our constructed benchmark dataset.

Table 1: Benchmark problem list of how-to instruction generation and story telling.

| Task | Problem |
| --- | --- |
| How-To Instructions | How to make monkey brains cocktail? |
| | How to make Korean banana milk? |
| | How to cook charcoal-grilled ribeye steak? |
| | How to make classic chocolate cupcakes? |
| | How to make ice-cream cake pops? |
| | How to make perfect gooey cinnamon rolls? |
| | How to do surfing? |
| | How to clean a glass bowl? |
| | How to trim a tree? |
| | How to cut a pineapple into pieces? |
| Story Telling | Can you write a story about the Spiderman? |
| | Can you write a story about the Iron Man? |
| | Can you write a story about the Superman? |
| | Can you write a story about the Batman? |
| | Can you write a story about the Hulk? |
| | Can you write a story about a white cat? |
| | Can you write a story about a giraffe? |
| | Can you write a story about a penguin? |
| | Can you write a story about a polar bear? |
| | Can you write a story about a moose? |

Table 2: Benchmark problem list of graphical story rewriting.

| Problem |
| --- |
| After flying a long distance, a thirsty crow was wandering the forest in search of water. Finally, he saw a pot half-filled with water. He tried to drink from it but his beak wasn't long enough to reach the water inside. He then saw pebbles on the ground and one by one, he put them in the pot until the water rose to the brim. The crow then hastily drank from it and quenched his thirst. |
| One day, a camel and her baby were chatting. The baby asked, "Mother, why do we have humps?" The mother replied, "Our humps are for storing water so that we can survive in the desert". "Oh", said the child, "and why do we have rounded feet mother?" "Because they are meant to help us walk comfortably in the desert. These legs help us move around in the sand." "Alright. But why are our eyelashes so long?" "To protect our eyes from the desert dust and sand. They are the protective covers for the eyes", replied the mother camel. The baby camel thought for a while and said, "So we have humps to store water for desert journeys, rounded hooves to keep us comfortable when we walk in the desert sand, and long eyelashes to protect us from sand and dust during a desert storm. Then what are we doing in a zoo?" The mother was dumbfounded. |
| One day, two goats try to cross a weak and narrow bridge across the river. The goats are at either end of the bridge, but neither is ready to make way for the other. They come to the centre of the bridge and begin fighting about who should cross first. As they fight mindlessly, the bridge gives in, taking both the goats down into the river with it. |
| Sitting on a lofty rock, an eagle was watching its prey move on the ground. A hunter, watching the eagle from behind a tree, shoots it with an arrow. As the eagle falls to the ground, with blood oozing from its wound, it sees that the arrow is made of its own plumage and thinks: "Alas, I am destroyed by an arrow made from my own feathers". |
| A fox sees a crow carrying a piece of cheese to a tree top. It decides to get the cheese for himself. It goes to the tree and starts praising the crow that it can sing better than a cuckoo. Hearing this, the crow beams with pride, and tries to sing. The piece of cheese falls to the ground as it opens its mouth to sing. The fox picks up the piece and runs away. |

Table 3: Benchmark problem list of webpage generation.

| Problem |
| --- |
| Title: Who is Arcteryx? Content: Arcteryx is a Canadian high-end design company specializing in outdoor apparel and equipment headquartered in North Vancouver, British Columbia. It focuses on technical apparel for mountaineering and Alpine sports, including related accessories. The name and logo of the company reference the Archaeopteryx, the transitional fossil of early dinosaurs to modern dinosaurs (birds). Arcteryx is known for its waterproof Gore-Tex shell jackets, knitwear, and down parkas. |
| Title: Who is Whole Foods? Content: Whole Foods Market, the largest American chain of supermarkets that specializes in natural and organic foods. It operates stores in the United States and also in Canada and the United Kingdom. Corporate headquarters are in Austin, Texas. In 2017 Whole Foods was acquired by Amazon.com. The first Whole Foods store opened its doors in Austin in September 1980, after John Mackey and Renee Lawson Hardy, owners of the SaferWay health food store, joined forces with Craig Weller and Mark Skiles, owners of Clarksville Natural Grocery. Somewhat larger than a typical health food store, it offered a wider selection of food. A flash flood tore through the uninsured building only a few months after the opening, but—with help from an already loyal core group of customers—the damage was quickly repaired. |
| Title: Who is Marvel Universe? Content: The Marvel Universe is a fictional shared universe where the stories in most American comic book titles and other media published by Marvel Comics take place. Super-teams such as the Avengers, the X-Men, the Fantastic Four, the Guardians of the Galaxy, and many Marvel superheroes live in this universe, including characters such as Spider-Man, Captain America, Iron Man, Thor, the Hulk, Ant-Man, the Wasp, Wolverine, Black Panther, Doctor Strange, Daredevil, and Captain Marvel, Blade, Black Widow, Hawkeye, among numerous others. It also contains well-known supervillains such as Doctor Doom, Magneto, Ultron, Thanos, Loki, The Green Goblin, Kang the Conqueror, Red Skull, The Kingpin, Doctor Octopus, Carnage, Apocalypse, Dormammu, Mysterio, Electro, and the Vulture. It also contains antiheroes such as Venom, Namor, Deadpool, Silver Sable, Ghost Rider, The Punisher, and Black Cat. |
| Title: Who is big agnes? Content: At our core, we want to inspire you to get outside and find the same appreciation for the backcountry we have. We are a small company with the mountains just minutes from our office. We test product in these mountains to make gear that you can trust when you are out there. Our passion is producing high quality outdoor equipment in a sustainable manner. We outfit all people with the gear needed to camp comfortably, explore the backcountry and have FUN! |
| Title: Who is Tesla, Inc.? Content: Tesla, Inc. is an American multinational automotive and clean energy company headquartered in Austin, Texas. Tesla designs and manufactures electric vehicles (cars and trucks), stationary battery energy storage devices from home to grid-scale, solar panels and solar shingles, and related products and services. Its subsidiary Tesla Energy develops and is a major installer of photovoltaic systems in the United States and is one of the largest global suppliers of battery energy storage systems with 6.5 gigawatt-hours (GWh) installed in 2022. |