# OpenReview forum: "OpenLEAF: Open-Domain Interleaved Image-Text Generation and Evaluation"
_ICLR.cc/2024/Conference — ICLR 2024 Conference Withdrawn Submission_

### Official Review · Reviewer_bkvJ · 2023-10-23

**Soundness:** 2 fair
**Presentation:** 2 fair
**Contribution:** 2 fair
**Rating:** 3
**Confidence:** 5

**Summary:**

The paper explores and proposes an interleaved generation framework based on prompting large-language models (LLMs) and pre-trained text-to-image (T2I) models. Additionally, it attempts to introduce an automated evaluation scheme based on LLMs for assessing Entity Consistency and Style Consistency.

**Strengths:**

This paper presents a interleaved generation framework based on prompting large-language models (LLMs) and pre-trained text-to-image (T2I) models, which includes user query composition, text generation, adding global context.

The paper also explore to present an LLM-based evaluation strategy for assessing interleaved content generation in two aspects: Entity Consistency Evaluation and Style Consistency Evaluation.

**Weaknesses:**

1. novelty.  The major concern with this paper is its excessive reliance on the APIs of existing models, GPT-4 and SDXL. The method appears more like a prompt engineering approach, devoid of the need for fine-tuning models, analyzing network structures, or delving into training strategies. It lacks the provision of novel insights, and from my perspective, this paper resembles more of a technical report than an academic contribution.

2. Entity Consistency and Style Consistency.  Indeed, Entity Consistency and Style Consistency are two crucial challenges in Storytelling generation. However, this work attempts to address these challenges merely using simple prompt description constraints, which appears to be a weak approach. Recently, some works[1] [2] [3] based on personalized LORA seem to offer a more reasonable solution for tackling consistency issues.

3. GPT4-version for evaluation. The proposed evaluation method appears to be quite rough and non-standardized. It seems to involve directly inputting the generated content into GPT-4 and asking whether it is reasonable or not, which is overly simplistic and engineering-oriented. Furthermore, the authors have not provided evidence for the validity of this method and have not offered new insights. At the very least, evaluations based on relevant personalized methods [1] [2] [3][4] would be necessary.

[1] Gu, Yuchao, Xintao Wang, Jay Zhangjie Wu, Yujun Shi, Yunpeng Chen, Zihan Fan, Wuyou Xiao et al. "Mix-of-Show: Decentralized Low-Rank Adaptation for Multi-Concept Customization of Diffusion Models." arXiv preprint arXiv:2305.18292 (2023).

[2] Chen, Xi, Lianghua Huang, Yu Liu, Yujun Shen, Deli Zhao, and Hengshuang Zhao. "AnyDoor: Zero-shot Object-level Image Customization." arXiv preprint arXiv:2307.09481 (2023).

[3] Guo, Yuwei, Ceyuan Yang, Anyi Rao, Yaohui Wang, Yu Qiao, Dahua Lin, and Bo Dai. "Animatediff: Animate your personalized text-to-image diffusion models without specific tuning." arXiv preprint arXiv:2307.04725 (2023).

[4] Ruiz, Nataniel, Yuanzhen Li, Varun Jampani, Yael Pritch, Michael Rubinstein, and Kfir Aberman. "Dreambooth: Fine tuning text-to-image diffusion models for subject-driven generation." In Proceedings of the IEEE/CVF Conference on Computer Vision and Pattern Recognition, pp. 22500-22510. 2023.

**Questions:**

None

---

### Official Review · Reviewer_d9kj · 2023-10-31

**Soundness:** 3 good
**Presentation:** 3 good
**Contribution:** 2 fair
**Rating:** 5
**Confidence:** 4

**Summary:**

This paper unveils the OpenLEAF framework, an inventive concoction aimed at catalyzing the synchronous generation of image-text content. OpenLEAF meticulously interweaves Large Language Models (LLMs) with Text-to-Image (T2I) models, crafting sequences enriched with coherence and quality. The inclusion of a benchmark dataset paired with a robust evaluation technique augments the paper’s appeal, cultivating a nuanced evaluative lens towards generated sequences.

**Strengths:**

(1) The paper presents a unified solution, offering a fresh perspective in the sphere of open-domain interleaved image-text generation.

(2) OpenLEAF emerges as a cohesive framework, harmonizing the strengths of LLMs and T2I models to birth sequences radiant with quality and coherence.

(3) The authors enrich the evaluation by incorporating a benchmark dataset and a fortified evaluation methodology, enhancing the objectivity and comprehensiveness of sequence evaluations.

**Weaknesses:**

(1) The paper's architectural foundation seems somewhat pre-ordained, leveraging predefined templates to navigate the realms of interleaved image-text generation. This approach echoes the contours of prompt engineering, utilizing well-established technical components like GPT-4 and SD-XL, making the technical contributions seem somewhat restrained and not profoundly innovative.

(2) A shadow of vulnerability seems to cloak the proposed mechanisms aimed at ensuring entity and style consistency. The reliance on a diffusion-based image generation model brews uncertainties regarding the model’s ability to consistently generate accurate and diversely styled results. The framework appears slightly handicapped in addressing or rectifying failures in such contexts, potentially moderating its utility.

(3) The paper’s experimental sections seem cluttered with multiple example cases, muddying the clarity of the presented insights. A thirst remains for a more enriched quantitative evaluation coupled with a nuanced exploration of failure scenarios to bolster the paper’s analytical depth.

(4) Some of the technical details are not clear. In User Query Composition, how to apply the Controls and ensure it would work (truly control the generated contents to follow such structure according the Controls). Is it possible that the model (GPT4) would not follow such templates to generate incorrect text results?

(5) There are minor typographical errors. For instance, inconsistencies like color discrepancies (blue "{" and black "}") in Fig. 2 (b) subtly detract from the visual clarity.

(6) The reliability of the BingChat evaluation in certain scenarios is doubtful. And a nuanced discussion or analytical dissection of these contexts seems conspicuously absent.

**Questions:**

N/A

---

### Official Review · Reviewer_vJS3 · 2023-11-01

**Soundness:** 1 poor
**Presentation:** 1 poor
**Contribution:** 2 fair
**Rating:** 3
**Confidence:** 4

**Summary:**

Open-domain interleaved image-text generation is the task where images and text are generated in an interleaved and coherent fashion. The paper introduces OpenLEAF by leveraging LLM (uses GPT-4) and T2I (uses SDXL) models. Generation instructions with examples are passed to GPT-4 for in-context learning, along with the user prompt and control sentences for counts.

The interleaved content between image tags in response is used by SDXL to generate an image. Global context is passed to the T2I model for appearance and style consistency. This task is natural for visual story telling.

Results (10 image pairs from generated interleaved results) are evaluated for consistency using BingChat whose effectiveness was validated through human evaluation and LLM analysis. The new benchmark contains 30 input queries spanning diverse tasks such as visual instruction generation (10), story generation (10), story rewriting (5), webpage/poster generation (5).

**Strengths:**

The problem is interesting and important. Solution is simple.

**Weaknesses:**

Experimental results are not comprehensive. Size of data is very small. No ablation studies. No comparison with baselines. This is really ad hoc prompt engineering for in-context learning (a corollary). The paper really highlights the power of GPT-4.

**Questions:**

Figure 1: Graphical Story Rewriting does not seem to have the right results (seems to be from a different story). Are these cherrypicked as well?

---

### Official Review · Reviewer_qwLm · 2023-11-04

**Soundness:** 3 good
**Presentation:** 3 good
**Contribution:** 2 fair
**Rating:** 5
**Confidence:** 4

**Summary:**

The paper presents a framework that leverages ChatGPT and SDXL to address open-domain interleaved image-text generation. Additionally, the paper investigates the ability of large multi-modal models to assess the consistency of entities and styles in open-domain interleaved image-text sequences. The experiments provide evidence of the effectiveness of the proposed framework and the evaluation capability of LMM.

**Strengths:**

1. The writing is clear and easy to follow.

2. The authors investigate the open-domain interleaved image-text generation task and explore the Language Model Metric's (LMM) assessment ability for this task.

3. The authors propose a framework for adopting existing models, such as ChatGPT and SDXL, to address the interleaved image-text generation task.

**Weaknesses:**

1. Basically, the paper introduces a framework for the interleaved image-text generation task, primarily by combining existing models, such as ChatGPT and SDXL. Given this, the method may not exhibit strong technical novelty.

2. The proposed method may require complex prompt generation to achieve the desired results, which might not be user-friendly for the average user.

**Questions:**

Please see above weaknesses.